

# Modeling the influence of temperature and water potential on seed germination of *Allium tenuissimum* L.

Hong Xiao[1], Helong Yang[1], Thomas Monaco[2], Qian Song[1] and Yuping Rong[1]

[1] Beijing Key Laboratory of Grassland Science, College of Grassland Science and Technology, China Agricultural University, Beijing, China
[2] USDA-ARS Forage and Range Research Laboratory, Utah State University, Logan, UT, United States of America

## ABSTRACT

*Allium tenuissimum* L. is a widely distributed perennial herbaceous species in temperate and desert steppes. Relative to other wild *Allium* species, it produces unique sweet flavors, more biomass in arid and cold environments, and has generated greater interest for crop production. Successful crop establishment, however, will depend on rapid and uniform seed germination. Our study aimed to characterize seed germination of *A. tenuissimum* under various temperature regimes (11, 15, 20, 24 and 28 °C) and water potential levels (0, −0.2, −0.4 and −0.6 MPa), and model germination by hydrotime (HT) and hydrothermal time (HTT) analysis. Final germination percentage (FGP) increased within the range of 11 to 20 °C, yet it declined within the range of 24 to 28 °C and generally decreased as water potential became more negative within each temperature setting. Maximum FGP was observed at 20 °C at all water potential settings and ranged from $55.0 \pm 5.3$ to $94.8 \pm 1.4$%. According to HT and HTT models, the base ($T_b$) and optimum temperatures ($T_o$) for seed germination were 7.0 and 20.5 °C, respectively. In addition, base water potential for the fraction of germination within the seed lot ($\Psi_b(g)$) shifted to 0 MPa as temperature increased from $T_b$ to ceiling temperature ($T_c$). For obtaining 50 % seed germination, $\Psi_b(50)$ and $T_c(50)$ were estimated to be −0.67 MPa and 27.2 °C, respectively. These values for $T_b$ and $\Psi_b(50)$ suggest seed germination of *A. tenuissimum* is both cold and drought tolerant and suitable for production in semi-arid regions. Our characterization of the ideal sowing conditions for *A. tenuissimum*, i.e., 20.5 °C and soil water potential less negative than −0.67 MPa offers information to forecast suitable settings to enhance crop production.

Corresponding author
Yuping Rong,
rongyuping@cau.edu.cn

## INTRODUCTION

Seed germination is an essential process in establishing stable plant populations and is regulated by many environmental factors (*Atashi et al., 2015*; *Juan-Vicedo et al., 2016*). Among these factors, temperature and moisture conditions strongly regulate germination dynamics (*Bakhshandeh & Gholamhossieni, 2019*), and characterizing the base, optimum, and ceiling temperatures is one way of describing how temperature influences seed germination dynamics (*Bakhshandeh et al., 2013*; *Bewley et al., 2013*). Furthermore,

thermal time (TT) models can be applied to predict seed germination dynamics across a range of temperatures (*Mccartan, Jinks & Barsoum, 2015*; *Bidgolya et al., 2018*; *Trudgill et al., 2005*) and have proven useful in estimating germination under the dynamic temperatures of field seedbeds (*Rawlins et al., 2012*; *Izquierdo et al., 2013*). However, TT models may inaccurately predict seed germination responses under supra-optimal temperature ranges (*Bakhshandeh et al., 2015*; *Bradford, 2002*).

Germination is also very sensitive to moisture conditions and water deficiency-induced osmotic stress is known to prevent seed germination or slow germination rates (*Tobe et al., 2001*). Accordingly, water status is typically reported as osmotic water potential, and hydrotime (HT) models are typically applied to simultaneously account for changes in both final germination percentage and germination rate across variable water potential levels (*Soltani et al., 2017*; *Bakhshandeh & Gholamhossieni, 2018*). Furthermore, the combination of HT and TT models have produced hydrothermal time (HTT) models (*Alvarado & Bradford, 2002*; *Gummerson, 1986*) capable of determining the hydrothermal accumulation for seed germination at various temperature and water potential conditions and predict the time course of seed germination even across sub-optimal (*Gummerson, 1986*) and supra-optimal temperature ranges (*Alvarado & Bradford, 2002*). Consequently, parameters of HTT models can be used to characterize the physiological status of seed populations in response to variable temperature and water potential and have been widely applied to predict germination dynamics in numerous crops such as safflower (Carthamus tinctorius) (*Bidgolya et al., 2018*; *Torabi et al., 2016*), sesame (*Sesamum indicum*) (*Bakhshandeh et al., 2017*), zucchini (*Cucurbita pepo*) (*Atashi et al., 2015*), and watermelon (*Citrullus vulgaris* cv. 'Crimson sweet') (*Bakhshandeh et al., 2015*) and other wild species (*Abdellaoui et al., 2019*; *Fakhfakh, Anjum & Chaieb, 2018*; *Horn, Nettles & Clair, 2015*).

*Allium tenuissimum* L. is a perennial herbaceous species distributed in temperate and desert steppes of north-central Asia (*He, 2008*; *Li & Zhang, 2011*). Within these ecosystems, it plays a critical role in sand fixation and conservation of water and soil due to having a well-developed fibrous root system and high tolerance of environmental stresses (*Zhao, 2010*). It is also recognized as a high-quality forage for herbivores (*Song, Niu & Wan, 2016*) and has economic importance due to its distinctive and tasty flavor (*Li & Zhang, 2011*; *Zhang & Liu, 2012*). Consequently, it is highly valued as a vegetable or food seasoning (*Li & Zhang, 2011*; *Liu et al., 2016*) and has generated interest in exploring new harvesting methods (*Zhang, Li & Zheng, 2014*), potential ways to extract essential oil from its flowers (*Zhang & Liu, 2012*), and identifying the unique volatile flavor compounds in flowers of *A. tenuissimum* (*Xu et al., 2017*). Despite these favorable qualities, establishing crops of *A. tenuissimum* on large-scales is not feasible by propagation of bulb tillers, but may become economically viable through propagation by seeds (*Li et al., 2013*; *Zhao, Li & Badema, 2011*). Therefore, exploring seed germination dynamics under variable environmental conditions is an important step toward industrialized production. In this study, we aimed to focus on the response of seed germination to various temperatures and water potential conditions in *A. tenuissimum*: (i) characterizing seed germination dynamics of *A. tenuissimum* using HT and HTT models, and (ii) defining cardinal temperatures and base water potentials for *A. tenuissimum* seeds based on model parameters.

## MATERIALS & METHODS

### Seed germination

Seeds of *A. teniussimum* were collected from a temperate desert steppe area in Dongsu County (43°51′36″N, 113°40′02″E, 1,060 m a.s.l. (meters above sea level)), Inner Mongolia during October 2015 (approved by Yuping Rong, China Agricultural University). After that, the seeds were naturally dried, surface cleaned and put into a sealed glass container, and then stored at 4 °C in a refrigerator until needed for experimentation (April 2016).

Germination assays were conducted in a growth chamber (SPX-250-GB, Hengyu, China) located at the Grassland Science Department, China Agricultural University, Beijing, China with an 8 h light/16 h dark day/night lighting pattern. The light intensity and relative humidity were set as 6000 lx and 50%. Mature Seeds were sterilized for 5 min with 10% NaClO and then washed with distilled water (*Rong, Li & Johnson, 2015*). For the germination tests, 100 seeds were placed in glass Petri dishes (90 mm inner diameter) containing two layers of filter paper (1001-090, Whatman, UK) saturated with distilled water ($\Psi = 0$ MPa) or solutions of different water potential levels ($\Psi = -0.2, -0.4$ and $-0.6$ MPa). Petri dishes were then transferred to the chamber and germination was characterized at four constant temperature settings (11, 15, 20, 24 and 28 °C), each replicated four times for each water potential level. Different water potential levels were produced by mixing aqueous solutions of polyethylene glycol (PEG) 6000 with distilled water according to *Michel & Kaufmann (1973)*. A vapour pressure osmometer (Model 5100C, Wescor, Inc., Logan, UT, USA) was used to measure $\Psi$ of solutions and create desired levels for all temperature settings. To maintain constant $\Psi$ and avoid fungal attack, seeds incubated on PEG 6000 solutions were transferred to fresh solutions every 2 d. Germination was scored daily by observing radicle protrusion. Seed germination was defined depend on the length of radicle. Normally seeds were regarded as germinated when the length of radicle was more than 2 mm (*Saleem et al., 2019*). To avoid errors in recording germination, germinated seeds were removed after being counted. Furthermore, germination tests were terminated when no new germinated seeds were counted for three consecutive days.

### Germination analysis

Germination rates ($GR_g$) were calculated using the equation: $GR_g = 1/t_g$, where $t_g$ is the duration to radicle emergence. Estimations of germination rate for the 50th percentile ($GR_{50}$) in each replicate were calculated by interpolation using curves fit to the time course data. To determine the optimal germination temperature, germination rates at 20 and 24 °C were compared. If final germination percentage at 20 °C was significantly greater than germination at 24 °C, the optimal temperature was assumed to be 20 °C.

### HT model

The relationship between $GR_g$ and $\Psi$ was described by using the following equation:

$$\theta_H = \left[\Psi - \Psi_b(g)\right] t_g \tag{1}$$

where $\theta_H$ is the hydrotime (MPa d) constant of the seeds required for germination, $\Psi_b(g)$ is the theoretical threshold or base $\Psi$ that will prevent the germination of fraction g. The

parameters in the HT model were estimated according to the equation:

$$\text{Probit(g)} = \left[ (\Psi - \theta_H / t_g) - \Psi_b(50) \right] / \sigma_{\Psi b} \tag{2}$$

where $\Psi_b(50)$ is the base water potential for attain the 50th percentile of germination, and $\sigma_{\Psi b}$ is the standard deviation of $\Psi_b(g)$ in seed lots. To examine whether parameters of this model can be accurately used to quantify the sensitivity of seed populations to the variation of $\Psi$, the $[1-(\Psi/\Psi(g))]$ $t_g$ factor derived from *Bradford (1990)* was applied to normalize germination time courses at reduced $\Psi$ in this study. Germination time course can be normalized by this factor at any $\Psi$ to the corresponding time course that would occur in water for the seed population using the parameters from the HT model. All normalized data from different temperature conditions were plotted on a common thermal time scale, using the estimated $T_b$ at 0 MPa.

## HTT model

The HTT model describes seed germination patterns when $T$ and $\Psi$ both vary; *Alvarado & Bradford, 2002*. The relationship between $GR_g$ and variable conditions of $T$ and $\Psi$ are described by Eq. (3) for sub-optimal $T$, and modified Eq. (4) for supra-optimal $T$:

$$\theta_{HT} = (T - T_b) \left[ \Psi - \Psi_b(g) \right] t_g \tag{3}$$

$$\theta_{HT} = \left\{ \Psi - \Psi_b(g) - \left[ K_T(T - T_o) \right] \right\} (T - T_b) t_g \tag{4}$$

where $\theta_{HT}$ is the hydrothermal constant and $T_b$ is the base temperature. In Eq. (4), $[K_T(T - T_o)]$ applies only in the supra-optimal range of $T$ and $K_T$ is the slope of the relationship between $\Psi_b(50)$ and temperatures when $T > T_o$; $T_o$ is the optimum temperature. The value of $\Psi_b(g)$ is set equal to the distribution of $\Psi_b(g)$ at $T_o$, and $(T - T_b)$ is equal to $(T_o - T_b)$. Parameter values in above models can be obtained by probit analysis according to Eqs. (5) and (6) for sub- and supra-optimal $T$, respectively.

$$\text{Probit(g)} = \left[ (\Psi - \theta_{HT} / (T - T_b) t_g) - \Psi_b(50) \right] / \sigma_{\Psi b} \tag{5}$$

$$\text{Probit(g)} = \left[ (\Psi - \theta_{HT} / (T - T_b) t_g) - K_T(T - T_o) - \Psi_b(50) \right] / \sigma_{\Psi b} \tag{6}$$

As described in *Alvarado & Bradford (2002)*, the values of $K_T$ and $T_o$ were varied for germination time courses at $T > T_o$ until a fit was obtained that resulted in values of $\theta_{HT}$, $\Psi_b(50)$ and $\sigma_{\Psi b}$ matching those obtained at or below $T_o$.

## Statistical analyses

A two-way analysis of variance (ANOVA) was carried out using SPSS 19.0 for Windows (SPSS Inc., Chicago, IL, USA) to evaluate the influence of $T$, $\Psi$, and their interactions on seed germination variables of *A. teniussimum*. The results showed with mean and SE value of four replicates. All probit analyses of HT and HTT models were fitted in SAS 8.2 statistical package (SAS Institute, Cary, NC, USA) using the PROC PROBIT routine, which employs a maximum-likelihood weighted regression method (*Bradford, 1990*; *Dahal & Bradford, 1994*).
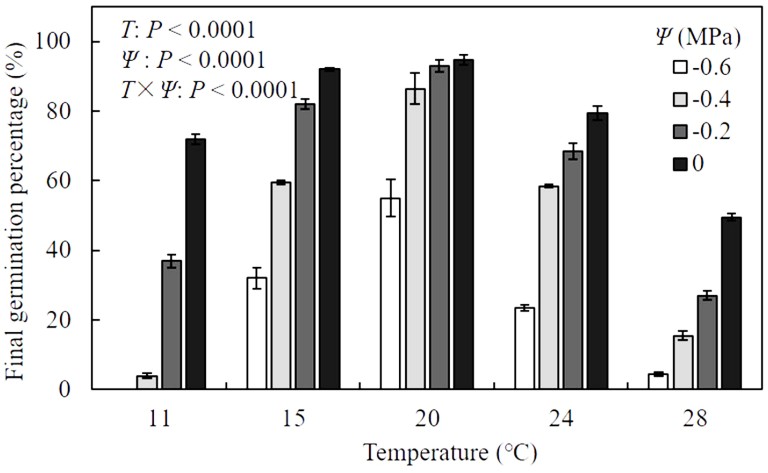

**Figure 1** Mean (± standard error) final germination percentage of *A. tenuissimum* seeds under variable temperature ($T = 11, 15, 20, 24,$ and $28\,°C$) and water potential ($\Psi = 0, -0.2, -0.4,$ and $-0.6$ MPa) levels.

## RESULTS

### Seed germination in response to temperature and water potential

Results of analysis of variance indicated that the final germination percentage (FGP) of *A. tenuissimum* seeds was significantly influenced by $T$ ($F = 587.9, P < 0.0001$), $\Psi$ ($F = 643.0, P < 0.0001$), and their interactions ($F = 24.2, P < 0.0001$) (Fig. 1, Table S1). When values of $\Psi$ remained constant, FGP increased as $T$ increased within sub-optimal ranges (11 to 20 °C), while it declined within supra-optimal ranges (24 to 28 °C). In distilled water (i.e., $\Psi = 0$ MPa), FGP changed from $72.0 \pm 1.4$ to $94.8 \pm 1.4\%$ over various $T$ conditions. Maximum FGP was observed at 20 °C under all levels of $\Psi$ and ranged from $55.0 \pm 5.3$ to $94.8 \pm 1.4\%$. For all $T$ settings, FGP decreased with decreasing $\Psi$ levels. Few seeds germinated under the combination of $-0.6$ MPa and 11 °C. In contrast, FGP reached $94.8 \pm 1.4\%$ when seeds were incubated in water at 20 °C, indicating that the seeds in this analysis were non-dormant.

### Hydrotime analysis

Parameters generated by the HT model are presented in Table 1. Values of $\theta_H$ decreased from 6.4 MPa d at 11 °C to 4.4, 4.2, 3.0 and 3.0 MPa d at 15, 20, 24 and 28 °C, respectively. This suggests that the time required for germination declined as $T$ increased within sub-optimal ranges, while it remained constant at supra-optimal ranges. Values of $\Psi_b(50)$ increased as $T$ increased from 11 to 28 °C. Notably, estimated values of $\Psi_b(50)$ increased more positively at supra-optimal temperatures, rising from $-0.40$ MPa at 24 °C to $-0.16$ MPa at 28 °C. Values of $\sigma_{\Psi b}$ varied from 0.20 to 0.37 MPa across all regimes of $T$. Specifically, values of $\sigma_{\Psi b}$ were nearly constant within the range of 15 to 24 °C, indicating that the variation in $\Psi_b$ among all seeds in this *A. tenuissimum* population was small.

**Table 1  Parameters of the hydrotime model characterizing germination of *A. tenuissimum* seeds across a range of temperatures.** The hydrotime time model (Eq. (2)) was fitted to data from four water potential levels (0, −0.2, −0.4 and −0.6 MPa) at five temperature regimes (11, 15, 20, 24 and 28 °C).

| Temperature (°C) | $\theta_H$ (MPa d) | $\Psi_b(50)$ (MPa) | $\sigma_{\Psi b}$ (MPa) | $R^2$ |
|---|---|---|---|---|
| 11 | 6.4 | −0.74 | 0.20 | 0.95 |
| 15 | 4.4 | −0.60 | 0.29 | 0.97 |
| 20 | 4.2 | −0.49 | 0.27 | 0.92 |
| 24 | 3.0 | −0.40 | 0.28 | 0.90 |
| 28 | 3.0 | −0.16 | 0.37 | 0.94 |

Notes.

$\theta_H$, hydrotime constant; $\Psi_b(50)$, base water potential for 50% seed germination; $\sigma_{\Psi b}$, standard deviation for $\Psi_b(g)$; $R^2$, coefficient of determination.

The curves in Figs. 2A–2E are germination time courses predicted by HT model based on the $\Psi_b(g)$ threshold distributions (Figs. 2F–2G) and the estimated parameters (Table 1). At each $T$ setting, the predicted curves closely matched actual seed germination data under the various $\Psi$ levels (Figs. 2A–2E). Normalization of germination time courses across various $\Psi$ levels at sub- and supra-optimal $T$ levels incorporated into a common curve are shown in Fig. 3. At sub-optimal $T$, the difference between groupings of normalized observations and common curve is indistinct (Fig. 3A). However, the grouping of normalized observations at 28 °C did not resemble the profile from the common curve and fell into a distinct group (Fig. 3B). This indicates that the estimates of HT model interacted with $T$. Furthermore, these HT estimates consistently showed the largest shift in $\Psi_b$ with increasing $T$ (Figs. 2F–2G) and the grouping of normalized observations accurately predicted seed germination in this population.

### Hydrothermal time analysis

Parameters for the HTT model, estimated in the sub- and supra-optimal $T$ ranges, are shown in Table 2. The hydrothermal time requirement ($\theta_{HT}$) for seed germination was 43.9 MPa °C d. Values $T_b$ and $\Psi_b(50)$ were estimated to be 7.0 °C and −0.67 MPa, respectively. FGP increased until 20.5 °C ($T_o$), then it declined towards ceiling temperatures ($T_c(g)$), at which germination theoretically ceased when $T$ exceeded $T_o$. In addition, $T_c(50)$, the ceiling temperature for germination of 50%, was 27.2 °C. The estimates for $K_T$ was 0.1 MPa °C$^{-1}$, indicating that $\Psi$ declined by 0.1 MPa for every degree that $T$ exceeds $T_o$. Applying the HTT models using the *A. tenuissimum* germination data resulted in high $R^2$ values at both sub-optimal ($R^2 = 0.89$) and supra-optimal ($R^2 = 0.81$) $T$ ranges, indicating a high degree of congruency between predicted and observed germination responses.

## DISCUSSION

### Seed germination of *A. tenuissimum* in response to various temperature and water availability conditions

Successful establishment of cultivated plants depends on rapid and uniform seed germination (*Fakhfakh, Anjum & Chaieb, 2018*; *Bidgolya et al., 2018*); however, suitable water availability and temperature conditions for seed germination are only available during a short period in most arid and semi-arid regions (*Fakhfakh, Anjum & Chaieb,*

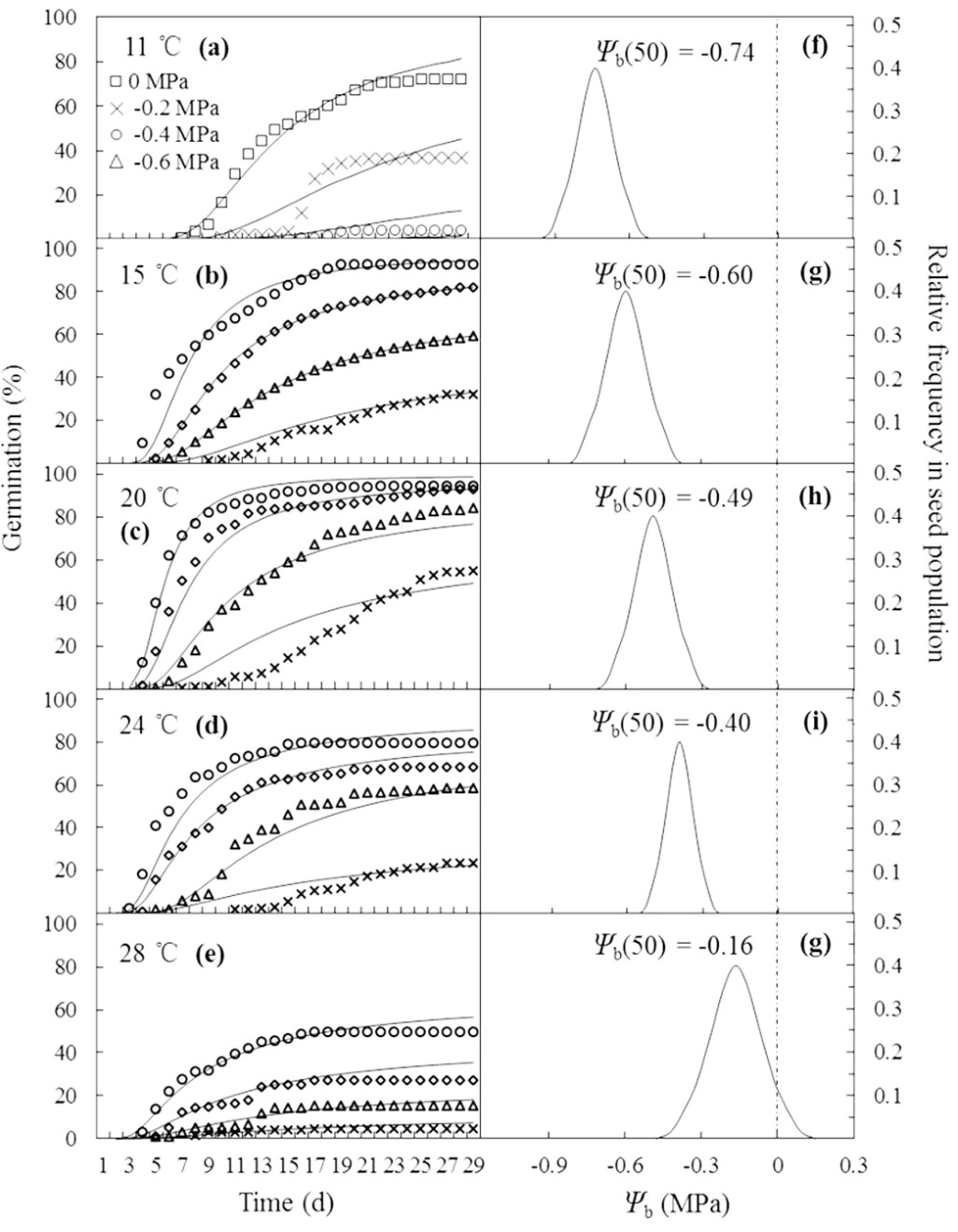

**Figure 2** Germination time courses of *A. tenuissimum* seeds across a range of water potential and temperature (A–E) and normal distributions of the relative frequencies of base water potential at each temperature (11, 15, 20, 24 and 28 °C) (F–G). Symbols represent actual data and lines indicate values predicted by probit analysis using the parameters presented in Table 1.

2018; *Belo et al., 2014*; *Watt & Bloomberg, 2012*). Here, we presented a comprehensive study of seed germination responses to different temperatures and water potential levels in *A. tenuissimum*, an important wild onion species from temperate-desert steppes of north central Asia (*He, 2008*; *Li & Zhang, 2011*). Our results showed that final germination
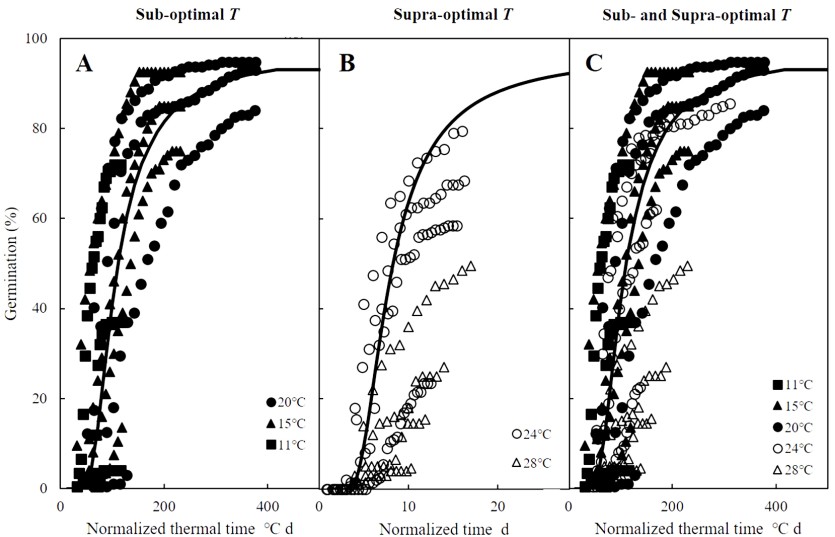

**Figure 3  Normalized time courses of the sub- and supra-optimal hydrotime models.** (A) Germination data from Figs. 2A–2C is plotted on a normalized thermal time scale showing the predicted time courses in water according to the parameters of sub-optimal $T$ shown in Table 2. (B) Germination data from Figs. 2D–2E are plotted on a normalized time scale showing the predicted time courses in water according to the parameters of supra-optimal $T$ shown in Table 2. (C) Germination time courses across all $T$ and $\Psi$ shown in Figs. 2A–2E merged into a single normalized thermal time scale.

**Table 2  Parameters of the hydrothermal time model characterizing germination of *A. tenuissimum* seeds at sub-optimal and supra-optimal $T$.**

| Hydrothermal time model parameters | Sub-optimal $T$ | Supra-optimal $T$ |
|---|---|---|
| $T_b$ (°C) | 7.0 | 7.0 |
| $\theta_{HT}$ (MPa °C d) | 43.9 | 43.9 |
| $\Psi_b(50)$ (MPa) | −0.67 | −0.67 |
| $\sigma_{\Psi b}$ (MPa) | 0.28 | 0.32 |
| $R^2$ | 0.89 | 0.81 |
| $T_o$ (°C) | – | 20.5 |
| $T_c(50)$ (°C) | – | 27.2 |
| $K_T$ (MPa °C$^{-1}$) | – | 0.1 |

**Notes.**

$T_b$, base temperature; $T_o$, optimal temperature; $\theta_{HT}$, hydrothermal time constant; $\Psi_b(50)$, base water potential for 50% seed germination; $\sigma_{\Psi b}$, standard deviation for $\Psi_b(g)$; $R^2$, coefficient of determination; $K_T$, the slope of the relationship between $\Psi_b(50)$ and temperatures when $T$ exceeds $T_o$; $T_c(50)$, ceiling temperature to germination of 50%. The value of $T_c(50)$ was calculated by parameters found after fitting the HTT model at supra-optimal $T$ (Eq. (6)).

percentage of *A. tenuissimum* strongly depended on the interaction of these two factors. Furthermore, applying HTT models to our dataset provided insights into suitable conditions to enhance consistent germination responses of this valuable native species. Later, we discuss in detail the nuances of our results that underpin this methodological approach and expedite industrialization of wild species under stressful environmental conditions.

Numerous studies have illustrated how temperature-dependent seed germination is related to the geographical and ecological distribution of a particular species such as common vetch (*Vicia sativa*) (*Liu, 2010*), *Stipa* species (*Ronnenberg, Wesche & Hensen, 2008*) and temperate sedges (*Carex*) (*Schütz & Rave, 1999*). For *A. tenuissimum*, the final germination percentage was quite high (i.e., 92.5 to 94.8%) in the temperature range of 15 to 20 °C (Fig. 1). In some mediterranean grassland species, 15 °C and 20 °C also were found to be the best temperature for seed germination (*Herranz, Ferrandis & Martínez-Sánchez, 1998*; *Marques & Draper, 2012*). In contrast, the lower temperatures for preferable germination of *A. tenuissimum* seeds was consistent with a prior study of wild species of *Allium* distributed in temperate desert steppe (*Zhao, Li & Badema, 2011*). In any constant temperatures, final germination percentage declined when seeds were incubated at reduced water potential levels (Fig. 1). This response is likely associated with enzyme activity and oxygen availability of seeds, which are known to decrease when germinated at unfavorable temperatures and limiting moisture conditions (*Bewley & Black, 1994*). Furthermore, higher incubation temperatures and more negative values of water potential during germination likely induced secondary dormancy of seeds, leading to prolonged seed germination over non-optimal temperatures and more negative water potential levels (Fig. 2).

## Predicting seed germination with mathematical models

Mathematical models to predict and quantify the influence of environmental factors on seed germination are essential when little information is known about the ideal conditions for potential agronomic species such as zucchini (*Cucurbita pepo*) (*Atashi et al., 2015*), red brome (*Bromus rubens*) and cheatgrass (*Bromus tectorum*) (*Horn, Nettles & Clair, 2015*), and safflowers (*Carthamus tinctorius*) (*Bidgolya et al., 2018*). While accounting for the influence of only temperature, TT models can successfully predict germination time courses in sub-optimal temperature ranges, but they become less effective at predicting germination in supra-optimal temperature ranges (*Bradford, 2002*; *Bewley et al., 2013*), which spurred the development of hydrothermal models combine the influences of temperature and water availability on seed germination (*Windauer et al., 2012*). Such HT and HTT models have been widely applied and accurately describe seed germination across variable temperatures and water potential levels (*Alvarado & Bradford, 2002*; *Bakhshandeh et al., 2015*; *Bakhshandeh et al., 2017*). In addition, normalizing thermal time scales, as we did for the germination time courses of *A. tenuissimum* (*Bradford, 1990*; *Bradford, 2002*), accurately described the influences of temperature and water availability (Fig. 3A). However, the grouping of normalized observations at 28 °C fell into a distinct group instead of resembling the profile from the common curve (Fig. 3B). Similarly, previous studies documented that HT analysis was unable to predict the germination time courses at supra-optimal temperatures (*Bakhshandeh et al., 2017*; *Rowse & Finch-Savage, 2010*), suggesting that HT estimates were not consistent across variable temperatures. In other words, grouping of normalized observations resulted in large shifts in the theoretical threshold or base water potential ($\Psi_b(g)$) that can prevent the fraction of germination as temperature increases (*Rong, Li & Johnson, 2015*). Furthermore, when the distribution of

$\Psi_b(g)$ overlaps with 0 MPa at a given temperature, germination of some seeds may be inhibited (*Alvarado & Bradford, 2002*; *Larsen et al., 2004*; *Watt, Bloomberg & Finch-Savage, 2011*). For this reason, *Alvarado & Bradford (2002)* developed the HTT model to account for the linear increase of $\Psi_b(g)$ value as temperature increased above $T_o$ in order to eliminate the positive shift in $\Psi_b(g)$ values at supra-optimal temperatures. This modified model can quantify and predict both final germination percentage and germination rate across different temperatures and water potential levels at which seed germination occurs (*Bakhshandeh et al., 2015*; *Rowse & Finch-Savage, 2010*) as we observed for germination time course of *A. tenuissimum* seeds at supra-optimal temperatures, which were well described by HTT model (i.e., $R^2 = 0.81$).

## Model parameters and their biological roles

Because changes in dormancy state are related to $\Psi_b$ (*Meyer, Debaenegill & Allen, 2000*) and high temperatures often induce secondary dormancy, soil water availability is critical for seed germination at supra-optimal temperatures (*Hills, Staden & Thomas, 2003*). Likewise, values of $\Psi_b(50)$ (−0.74 to −0.16 MPa) determined in our analysis revealed that seeds of *A. tenuissimum* are relatively sensitive to water restrictions at temperatures above $T_o$. Thus, seeds of *A. tenuissimum* should be germinated at sub-optimal temperatures to enhance crop establishment in arid and semi-arid regions. Results from HT analysis also indicated that the minimum value of $\Psi_b(50)$ was observed at 11 °C and then increased, particularly in supra-optimal temperature ranges. Similar to our results, the linear increase of $\Psi_b(50)$ has been observed in zucchini (*Atashi et al., 2015*), watermelon (*Bakhshandeh et al., 2015*), and sesame (*Bakhshandeh et al., 2017*). In general, the $\Psi_b(50)$ value of a seed lot gives an indication of its tolerance to water stress. If water potential levels are more negative than $\Psi_b(50)$, germination times will be extended and germination rates will be reduced (*Bradford, 2002*). This could be caused by a decrease in both enzyme activity and oxygen availability during the seed germination period, particularly when germinated at supra-optimal temperatures (*Bewley et al., 2013*). When soil temperature approaches $T_o$, less negative water potential levels will cause an increase in the activity of enzymes and water uptake rate (*Kebreab & Murdoch, 1999*).

Base water potential coefficient ($\sigma_{\Psi b}$) is related to life history strategy of various species (*Bradford, 1990*) and indicates the uniformity of seed germination among individual seeds within a seed lot. A smaller value of $\sigma_{\Psi b}$ represents an increasingly uniform germination among seed population (*Bradford & Still, 2004*; *Bidgolya et al., 2018*). The values of $\sigma_{\Psi b}$ we estimated for *A. tenuissimum* seeds varied from 0.20 to 0.37 MPa, indicating that there was less variation in $\Psi_b$ among individual seeds and that uniform germination was observed. This pattern is likely a reflection of survival adaptations to harsh environments. Under initial favorable soil temperature and water availability, faster and uniform germination will allow plants to possibly dominate a plant community in space and time (*Wang et al., 2009*). In contrast, most plant populations must adopt a different germination strategy to mitigate the harsh environment-induced damage on germination by allowing a few seeds to rapidly germinate, and then delaying germination until consistent suitable environment conditions are reached before the remaining seeds germinate (*Bradford, 1990*; *Batlla et al.,*

*2009*; *Watt, Bloomberg & Finch-Savage, 2011*; *Rong, Li & Johnson, 2015*). Consistent with our results, the variation of $\Psi_b$ was small and $\sigma_{\Psi b}$ varied from 0.16 to 0.31 MPa among *Vicia sativa* seeds (*Liu, 2010*), indicating that the long-term cultivation and domestication can result in uniform germination.

The hydrotime (MPa d) constant of the seeds required for germination ($\theta_H$) represents the inherent speed of seed germination in a seed lot (*Bradford & Still, 2004*). Our results showed that $\theta_H$ declined from 6.4 to 3.0 MPa d as temperature increased from 11 °C to 24 °C, indicating that germination rate was faster at higher temperatures (Fig. 2, Table 1). Similarly, other studies reported that $\theta_H$ was constant in supra-optimal temperature ranges, but it increased as temperature declined in the range of sub-optimal temperatures (*Bakhshandeh et al., 2015*; *Dahal & Bradford, 1994*). Thus, seed germination was greatly altered with decreasing temperature.

Seed germination responses to temperature are commonly characterized with three cardinal temperatures ($T_b$, $T_o$ and $T_c$) estimated according to the HTT model (*Bewley et al., 2013*). These cardinal temperatures of *A. tenuissimum* seeds were $T_b = 7.0$ °C, $T_o = 20.5$ °C, and $T_c(50) = 27.2$ °C, respectively. In practical terms, these values suggest *A. tenuissimum* seeds should not be sown in soils where temperature do not exceed 7.0 °C. Maximum final germination percentage would be observed as soil temperature approaches 20.5 °C, a value when some seeds within the population will probably germinate quickly if they are not exposed to water stress or dormancy induction within optimal ranges (*Rong, Li & Johnson, 2015*; *Windauer et al., 2012*). In addition, because seeds of *A. tenuissimum* germinated in a relatively narrow range of temperature according to estimations of $T_c(50)$, seeds of *A. tenuissimum* are likely unable to tolerate high soil temperature. From a crop production standpoint, our results document that seed germination of *A. tenuissimum* occurs over a soil temperature range of 7.0 to 27.2 °C, but 20.5 °C is the optimal temperature. These estimated cardinal temperatures are consistent with wild *Allium* species (*Zhao, Li & Badema, 2011*).

## CONCLUSIONS

The mathematical models or the estimated value of parameters can be used to quantitatively predict the seed germination of *A. tenuissimum* under various T and $\Psi$ conditions. Soil temperature for seed germination of *A. tenuissimum* should be at the range of 7.0 °C to 27.2 °C, and the optimum temperature is 20.5 °C. The water potential should be less negative than −0.67 MPa. The relatively lower $T_b$ and more negative $\Psi_b(50)$ indicate that this wild *Allium* species might be established in arid and semi-arid regions, while relatively narrow threshold in response to T and $\Psi$ variations might sufficiently delay or even prevent seed germination in extreme arid or harsh desert regions.

### Funding

This research was supported by the National Key Research and Development Program of China (No. 2016YFC0500603), the National Natural Science Foundation of China (No. 31772653) and National Natural Science Foundation of China (No. 31472136). The funders had no role in study design, data collection and analysis, decision to publish, or preparation of the manuscript.

### Grant Disclosures

The following grant information was disclosed by the authors:
National Key Research and Development Program of China: 2016YFC0500603.
National Natural Science Foundation of China: 31772653, 31472136.

### Competing Interests

The authors declare there are no competing interests.

### Author Contributions

- Hong Xiao conceived and designed the experiments, performed the experiments, analyzed the data, prepared figures and/or tables, authored or reviewed drafts of the paper, and approved the final draft.
- Helong Yang performed the experiments, analyzed the data, prepared figures and/or tables, and approved the final draft.
- Thomas Monaco conceived and designed the experiments, authored or reviewed drafts of the paper, and approved the final draft.
- Qian Song performed the experiments, prepared figures and/or tables, and approved the final draft.
- Yuping Rong conceived and designed the experiments, analyzed the data, authored or reviewed drafts of the paper, and approved the final draft.

### Field Study Permissions

The following information was supplied relating to field study approvals (i.e., approving body and any reference numbers):

Shounan Shen and Wenxi Sun approved access to the study sites prior to seed collection. Shounan Shen and Wenxi Sun own the land and work the land as herdsmen at the study sites where we collected our seed samples. Shizhong Sun, the village headman who oversees the study area, approved access to the study sites on behalf of residents of the closest village to the study site.

### Data Availability

Raw data is available in the Supplemental Files.

## Supplemental Information

Supplemental information for this article can be found online at http://dx.doi.org/10.7717/peerj.8866#supplemental-information.

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
