# Peer review of "Modeling the influence of temperature and water potential on seed germination of Allium tenuissimum L"

_PeerJ, doi:10.7717/peerj.8866_

## Round 0.1 · original submission · Major Revisions

Your submission has been reviewed by experts in the field and we request that you make major revisions before it is processed further.

Reviewer 1 ·

Basic reporting

The manuscript is overall easy to read and to understand. However, there are still some minor typos that could be fixed. For instance in the abstract the authors say "Our study aimed to characterize seed germination of A. tenuissimum under various temperature (11, 15, 20, 24 and 28 °C)"; it should be under various temperatures (plural instead of singular) or even "under various temperature regimes".
This is only an example but it occurs multiple times throughout the text. However, I consider this to be minor, and can be easily fixed.

Figures and tables are fine but I missed the statistical information on them. For instance, post-hoc tests can be easily incorporated on Figure 1 (also on the text). F values and d.f are missing in the text.

Literature references can be largely improved (as well as the Discussion). Several articles also found 15C and 20C to be the best temperature for seed germination. See for instance:
https://www.ingentaconnect.com/content/ista/sst/2016/00000044/00000001/art00017
https://www.researchgate.net/profile/David_Draper_Munt/publication/233883683_Germination_behaviour_of_seven_mediterranean_grassland_species/links/0deec52e9227a54a59000000.pdf
https://www.cambridge.org/core/journals/seed-science-research/article/seed-germination-and-longevity-of-autumnflowering-and-autumnseed-producing-mediterranean-geophytes/D6BFFAB2EA18201F722DA29998EFCA76
https://www.researchgate.net/publication/227123219_Seed_germination_of_high_mountain_Mediterranean_species_Altitudinal_interpopulation_and_interannual_variability

Experimental design

Models need to be explained with more details, as well as germination tests. For instance:
- were seeds dried before used?
- they were kept in a sealed glass. That is not a good procedure to store seeds as it usually led to the loss of viability. How has this affected the results?
- were seeds kept under light? What about during germination?
- I assumed that results of germination are the mean and SE (or SD?) of the four replicates. This needs to be explained.
- Was seed viability of non-germinated seeds checked?
- Final germination in % usually needs an arc-sin transformation before statistical analysis. Was this done?

Validity of the findings

Overall conclusions seem to be well stated but there is some confusion between the parameters measured. This needs to be better explained, although it probably might not affect conclusions.
As a major point, the results found and the discussion need to be better explained in light of previous results.

Additional comments

A main objective of this paper was to characterize seed germination under variable temperatures and water potential to define parameters that can be used in industrialized production. How exactly can the industry implement these findings in crop production?

Reviewer 2 ·

Basic reporting

The manuscript entitled “Modeling the influence of temperature and water potential on seed germination of Allium tenuissimum” has noval information about model germination by hydrotime and hydrothermal time analysis. Under different temperatures and water potential the growth or seed germination was most suitable in cold and drought conditions is 20.5 °C. I suggest this manuscript should be publish in Peer J after Some major concerns.

Experimental design

The design looks satisfactory although I have some objections?
The Author try to find a suitable temperature for the growth of Allium tenuissimum and used various temperatures (11, 15, 20, 23 24 and 28 °C) is these temperatures are very low? Why should you not tried with 35 °C or may be 40 °C. Although, this temperature is common in various part of the world. Please explain it briefly?

Validity of the findings

The results from the current study is meaningful.

Additional comments

Before accept, several changes needed to be done.
1. Throughout your manuscript you just used “Allium tenuissimum” Even in the title you did not mentioned it Author name? please write its common name also even in text or provide taxonomic name.
2. L31 in abstract do not provide space in (50 %).
3. Introduction L 42 can you cite latest reference?
4. Introduction L 45-46 references looks too old for this important line.
5. What is the relationship between water potential and supra-optimal temperature ranges?
6. Introduction L 54-55 references again looks too old.
7. Introduction L 62. Better to write the refences with the crop name. then it will be easy to understand.
8. Introduction L 68: please make sure that Allium tenuissimum’s roots are fibrous?
9. “has economic importance due to its distinctive and tasty flavour” cite reference here
10. “exploration of its unique volatile compounds” difficult to understand.
11. I suggest you to write your objectives in more effective ways. Like In this study we aimed to focus on Allium tenuissimum under different environment condition (i), (ii) and (ii). Please make points and write it again then it will be clearer to understand your study.
12. Materials & methods L 84: unable to understand the meaning of m a.s.l. remove it or define it.
13. Materials & methods L 85: Author has mentioned “Germination assays were conducted in a growth chamber located at the Grassland Science Department, China Agricultural University, Beijing, China with an 8 h 87 light/16 h dark day/night lighting pattern”. Here I suggest you to add some more information. Like about growth chamber? What about humidity factor? And temperature was constant?
14. “Seeds were sterilized for 5 min with 10 % NaClO and then washed with distilled water” please cite some reference.
15. “Seeds were sterilized for 5 min with 10 % NaClO and then washed with distilled water” Also can mention mature seeds?
16. Please add some detail about filter paper?
17. Why used two layers of filter paper?
18. “solutions were transferred to fresh solutions every 2 d” to also avoid fungal attack?
19. “Germination was scored daily by observing radicle protrusion”. When seedlings get germinated or should mention the definition of seed germination. Normally Seeds were germinated when radicle was more than 2 mm. As mention in this article. doi:10.3390/plants8120545. Please verify from this article and cite in this sentence.
20. L 194: Reference looks too old.
21. L206-207: Please write your reference with crop names as earlier mention.
22. “yet there was a poor fit at 28 °C” difficult to understand.
23. L 223: Same comment, write the crop names?
24. At 0 MPa Germination was inhibited? Any logic or reference?
25. I just suggest you please write your conclusions simply as you can. Means it should be general.

---

## Round 0.2 · accepted · Accept

The authors provide an example for modeling seed germination to enhance crop production. The improper request for literature citation can be ignored.

Reviewer 2 ·

Basic reporting

The Author have made the changes according to the given suggestions. However, I request to Author please cite the following articles in the introduction section: https://doi.org/10.1016/j.chemosphere.2020.126032
https://doi.org/10.1016/j.ecoenv.2019.109915
https://doi.org/10.1016/j.jenvman.2019.109994

Experimental design

No comments

Validity of the findings

No comments

Additional comments

No comments